# Tissue-Specific Transcriptome Analysis Reveals Candidate Transcripts Associated with the Process of Programmed B Chromosome Elimination in *Aegilops speltoides*

**DOI:** 10.3390/ijms21207596

**Published:** 2020-10-14

**Authors:** Anastassia Boudichevskaia, Alevtina Ruban, Johannes Thiel, Anne Fiebig, Andreas Houben

**Affiliations:** 1Leibniz Institute of Plant Genetics and Crop Plant Research (IPK) Gatersleben, Seeland, 06466 OT Gatersleben, Germany; alevtina.ruban@kws.com (A.R.); thielj@ipk-gatersleben.de (J.T.); fiebig@ipk-gatersleben.de (A.F.); 2KWS SAAT SE & Co. KGaA, 37574 Einbeck, Germany

**Keywords:** chromosome elimination, RNA-seq, B chromosome, programmed DNA elimination

## Abstract

Some eukaryotes exhibit dramatic genome size differences between cells of different organs, resulting from programmed elimination of chromosomes. Here, we present the first transcriptome analysis of programmed chromosome elimination using laser capture microdissection (LCM)-based isolation of the central meristematic region of *Aegilops speltoides* embryos where B chromosome (B) elimination occurs. The comparative RNA-seq analysis of meristematic cells of embryos with (Bplus) and without Bs (B0) allowed the identification of 14,578 transcript isoforms (35% out of 41,615 analyzed transcript isoforms) that are differentially expressed during the elimination of Bs. A total of 2908 annotated unigenes were found to be up-regulated in Bplus condition. These genes are either associated with the process of B chromosome elimination or with the presence of B chromosomes themselves. GO enrichment analysis categorized up-regulated transcript isoforms into 27 overrepresented terms related to the biological process, nine terms of the molecular function aspect and three terms of the cellular component category. A total of 2726 annotated unigenes were down-regulated in Bplus condition. Based on strict filtering criteria, 341 B-unique transcript isoforms could be identified in central meristematic cells, of which 70 were functionally annotated. Beside others, genes associated with chromosome segregation, kinetochore function and spindle checkpoint activity were retrieved as promising candidates involved in the process of B chromosome elimination.

## 1. Introduction

The genetic information is believed to be uniform in somatic and gamete precursor cells in most organisms. However, some species demonstrate regular elimination of specific DNA fragments as a part of the developmental program. This occurs either as a loss of entire chromosomes, chromosome fragments or DNA sequences, often during the differentiation of somatic and germline cells. The so-called programmed DNA elimination takes place in the ontogenesis of some unicellular ciliates and diverse metazoa species (reviewed by [1,2]).

In plants, only B chromosomes are known to be associated with programmed DNA elimination [3]. B chromosomes (Bs) can be observed in some plant, animal and fungi species as a dispensable addition to the basic karyotype. Their presence may be neutral in regard to the phenotypic characteristics of the host or it may exert detrimental effects on the fitness and fertility, often depending on the number of Bs in individuals [4]. Analysis of different species demonstrated that Bs encode active genic sequences and also influence the transcriptome of standard genomes, reviewed by [5]. In some plant species, such as *Sorghum purpureosericeum*, *Xanthisma texanum*, *Agropyron cristatum*, *Poa alpina*, *Aegilops mutica* and *Aegilops speltoides*, distribution of Bs was shown to be organ-specific [6,7,8,9,10,11]. In B-carrying individuals of these species, Bs are completely absent in root cells, while they can be observed in cells of other organs.

The goat grass *Aegilops speltoides* may possess up to eight B chromosomes in individual plants, which are stably present in the cells of all organs except in roots [6]. An effect of Bs on the phenotype is visible when the number of Bs exceeds four (Figure 1). The root-restricted elimination of Bs starts with the formation of the radicle, seven–eight days after pollination, at the onset of embryo differentiation and proceeds until embryo maturation. The elimination of Bs is triggered by the centromere activity-independent nondisjunction of B-chromatids during the anaphase of proto-root cells. Subsequently, Bs undergo micronucleation and complete degradation. Although some B chromosome-specific genes of *Ae. speltoides* were identified [3], no information about genes involved in chromosome elimination exists. Moreover, it is not known whether this process affects the transcriptome of developing embryos in an indirect way.

To evaluate the impact of B chromosome elimination on the embryonic transcriptome and to identify candidate transcripts associated with the process of B elimination, we conducted a comparative analysis of central meristematic regions of B-containing embryos and embryos without Bs. Tissue-specific RNA-seq revealed that the elimination of Bs has a strong effect on the transcriptome of the meristematic central zone of young embryos. In particular, 35% from 41,615 analyzed transcript isoforms showed statistically significant altered expression during the elimination process. These transcripts are either associated with the process of B chromosome elimination, transcribed by Bs irrespectively of the elimination process or transcribed by the standard A chromosomes in response to the presence of Bs. Genes associated with chromosome segregation stand out as candidates involved in the process of B chromosome elimination.

## 2. Results and Discussion

### 2.1. The Transcriptome of Embryos Undergoing B Chromosome Elimination Is Strongly Affected

Elimination of *A. speltodies* B chromosomes occurs in the central meristematic zone between the developing apical meristem and embryonic root [3]. To evaluate whether the process of B chromosome elimination affects the transcriptome of developing embryos, the central meristematic regions of 17 to 20 days after anthesis (DAA) embryos of mother plants with (called Bplus) and without B chromosomes (called B0) were isolated by LCM (Figure 2), and RNA-seq was performed for transcriptome profiling. 

A de novo transcriptome assembly based on Trinity was applied since *Ae. speltoides* represents a non-model organism that contains in addition a high proportion of heterozygous loci. Trinity is highly effective for reconstructing transcripts and alternative spliced isoforms [12], and it allowed a better understanding of which transcript isoforms are truly expressed under a specific condition such as elimination of Bs. The de novo transcriptome assembly resulted in B0 and Bplus FASTA files containing 165,818 and 196,360 contigs, respectively. Subsequent filtering resulted in 117,441 contigs for downstream analysis (N50 contig size of 1026 bp). Detailed information about the assembly statistics is given in Appendix A. Transcriptome profiles of *Ae. speltoides* were evaluated at the transcript isoform expression level and quality was checked with the script PtR. A correlation matrix and PCA confirmed biological replications (Figure 3, Appendix A).

Differential expression analysis identified 14,578 transcript isoforms that were differentially expressed during B elimination (FC of log_2_ > 1, FDR-corrected *p*-value < 0.01, Table 1, Figure 4). Transcript isoforms significantly changing their expression during B chromosome elimination can either reflect the expression from B-located genic sequences or represent a response of A-located genes as a consequence of B elimination. 

The high number of differentially regulated transcript isoforms (35% out of 41,615 analyzed transcript isoforms) indicates that the process of B chromosome elimination and the presence of B chromosomes have a strong effect on the transcriptome of the embryonic central meristematic region (Figure 4). After stringent filtering, we could define 502 B0-specific (normalized mean read amount in Bplus is 0) and 341 Bplus-specific (normalized mean read amount in B0 is 0) transcript isoforms. In other cases, due to the high sequence similarity of A and B chromosome-located coding sequences [3], it was not possible to distinguish the origin of transcripts.

To verify RNA-seq data, reverse transcription quantitative PCR (RT-qPCR) was performed with a subset of transcripts known to be involved in cytokinesis and microtubule organization, similarly found to be B0- or Bplus-specific (Figure 5). The kinetochore gene *Nuf2* (nuclear filament-containing protein), Bplus-specific *BREVIS RADIX* (*BRX*), a regulator of cell proliferation and elongation in the root and shoot, *SMC2*, a component of the chromosome condensin complex, and *TPX2*, encoding a Xklp2-targeting protein, showed nearly no expression in the meristematic region of B0 embryos. Preferential expression in that cellular region of Bplus embryos was also shown for the chromosome segregation protein SPC25, whereas B0-specific expression was confirmed for a different isoform of *BRX*. Transcript levels and similar expression differences between candidate transcripts obtained by RT-qPCR validated the quality of the RNA-seq data analysis.

### 2.2. Biological Processes Potentially Related to the Elimination of B Chromosomes

The final data set containing non-redundant B0 and Bplus transcripts (117,441 contigs in total) was screened using Transdecoder v5.3.0 to predict open reading frames and resulted in 61,817 protein-coding transcripts. Such a high number of transcripts is in line with previous findings [13], where a global gene expression atlas of developing embryos of wheat and their putative diploid ancestors including *Ae. speltoides* was provided.

The completeness of the transcriptome assembly was assessed by BUSCO. Detailed results are displayed in Appendix A. The level of transcriptome completeness depends on the amount of tissues and developmental stages within the samples [14]. Therefore, we would expect a limited set of detected BUSCOs in our tissue-specific analysis of meristematic embryo cells. Nevertheless, we were able to recover a total fraction of ~65% BUSCO-confirmed genes (including fragmented and complete). Comparing our findings to a recently published study in an oat embryonic transcriptome detecting 60% complete and fragmented orthologues [15], the BUSCO results for the tissue-specific *Ae. speltoides* transcriptome are highly similar.

The resulting protein-coding transcripts were annotated based on Interproscan Galaxy [16,17] and 26,722 transcript isoforms with known functions could be obtained, which are represented in 23,836 unigenes (threshold of 1.0 × 10^−3^). The differentially expressed transcripts were aligned [18] against the hexaploid wheat genome assembly (IWGSC Ref.Seq.1.0, HighConf_Protein_2017), and transcripts were selected using the first best hit criterion with a minimum length of matching sequences of 100 amino acids. As a result, 8415 differentially expressed transcript sequences were annotated using proteins of the wheat genome. 

To determine overrepresented biological processes associated with the elimination of the B chromosomes, the Gene Set Enrichment Analysis tool (GSEA, AgriGo v2. analytical toolkit) with wheat as a background reference was used. Separated analyses of up- and down-regulated transcript isoforms undergoing B chromosome elimination were performed. The results revealed different enriched GO terms associated with down- and up-regulated transcript isoforms.

The dataset of up-regulated transcripts in meristematic regions of B-containing embryos (Bplus) contains 2908 annotated wheat genes. Among them, 27 terms for biological processes, nine terms for molecular function and three terms for cellular component were overrepresented (FDR-corrected *p*-value < 0.05, Appendix A). In the category of biological processes, “cellular process” (989, 34.0% from dataset) and “nitrogen compound metabolic process” (467, 16.1%) were highly enriched. Other overrepresented terms were “gene expression”, “microtubule-based process”, “movement of cell or subcellular component”, “microtubule-based movement” and “organelle fission”. Terms in the molecular function category, particular “guanyl nucleotide binding” and “GTP binding”, but also other enriched categories, such as “cytoskeletal protein binding”, “microtubule binding”, “microtubule motor activity” and “tubulin binding” coincide with the biological process category and additionally highlight that cytoskeleton/microtubule organization and vesicle trafficking are mostly affected in cells undergoing B elimination. In the cellular component category, only three terms, namely “cytoplasm”, “chromosome, centromeric region” and “chromosomal region”, were enriched. The results of GO enrichment are depicted in Figure 6, Appendix A and Appendix A (Appendix A). Similar enriched GO terms (“cellular process”, “microtubule-based movement”, “cell division”) were identified by [19] in the analysis of rye B transcripts of leaf and anther tissues. 

Some kinesin-like proteins are up-regulated in Bplus embryos and are deemed to be involved in microtubule binding, movement and motor activity (Appendix A). Furthermore, a transcript (Bplus_DN37268_c0_g1_i1) encoding Katanin p60 ATPase-containing subunit A-like 1 (p60 katanin-like 1) was highly up-regulated in the Bplus condition (log_2_ FC of 6.16, FDR-corrected *p*-value of 0.0009). Katanin is the only defined microtubule severing protein affecting microtubule organization in higher plants [20].

Three transcript isoforms of the *Nuf2* gene, which is crucial for chromosome segregation and spindle checkpoint activity, were significantly up-regulated (log_2_ FC value > 1, FDR-corrected *p*-value < 0.05) in Bplus samples (Appendix A), of which isoform c_Bplus_DN38911_c1_g1_i9 shows the highest specificity for Bplus condition (see also Figure 5). *Nuf2* is evolutionary conserved and known to play a role in spindle checkpoint control by regulating the bipolar attachment of microtubules of the sister chromatids before and after anaphase. Overexpression of *Nuf2* in mammals also caused defects in chromosome segregation [21]. Hence, as deregulation of *Nuf2* expression results in chromosomal segregation defects, the balanced activity of *Nuf2* is essential for regular chromosome segregation. In the context of the observed B-specific overexpression of *Nuf2* in embryos undergoing chromosome elimination, it is tempting to speculate that overexpression per se or expression of B-specific *Nuf2* variants is part of the observed B chromosome segregation alterations. Further transcripts encoding the spindle checkpoint controlling component Shugoshin (Sgo1), a Mis12 protein that is involved in kinetochore formation (Appendix A), and the spindle and kinetochore-associated protein 2 (Ska2) are highly activated in cells undergoing B chromosome elimination. 

The dataset of down-regulated transcripts in meristematic regions of B-containing embryos is represented with 2726 annotated genes. The category biological processes included 38 enriched terms, such as “DNA repair”, “methylation” and “compound nitrogen metabolic process”. In the category molecular function, only “oligosaccharyl transferase activity” was significantly enriched. The category cellular component includes 16 enriched terms with the most prominent terms “cell” and “cell part” (Appendix A). Gene set enrichment analysis demonstrated extensive modulation of genes involved in stress response (term “cellular response to stress”, Appendix A) during the process of B chromosome elimination. For example, a gene encoding an important tumor suppressor, *BRCA1*, was highly down-regulated in the Bplus condition. *BRCA1*, known as a regulator of crucial cellular processes, protects cells from aneuploidy and genomic instability. Its down-regulation promotes aberrant mitoses and aneuploidy and thereby it is a hallmark of cancer [22]. Further, genes encoding proteins such as DNA repair protein RAD51 (RAD51D), DNA repair helicase, DNA mismatch repair protein and DNA damage checkpoint protein Rad9 were highly down-regulated in meristematic cells undergoing B chromosome elimination indicating that chromosomal repair mechanisms might be defected. 

### 2.3. B Chromosome-Specific Transcripts

Based on strict filtering criteria, 341 B-unique transcript isoforms were identified and 70 of them were annotated (Appendix A). Transcript isoforms detected during chromosome elimination represented several categories: transposons (such as transposase, retrotransposon); potential transposons (zinc finger); unknown; and genes with specified functions. Among genes with specified functions, there were two genes encoding kinesin-like proteins. One full-length protein showed 94% identity to the kinesin-like protein KIN-14C of *Ae. tauschii* subsp *tauschii*. In *A. thaliana*, KIN_14C is important for regular microtubule accumulation at the spindle poles during the prophase of mitosis. Interestingly, B chromosome-specific genes related to the microtubules and cell division were also identified in several species [5].

The reason why B chromosomes undergo elimination in roots is unknown. However, it was suggested that elimination of Bs counteracts negative effects potentially associated with the fitness and fertility of the plants [3]. Our tissue-specific study revealed that some of the B chromosome-specific transcript isoforms encode regulators of plant growth and development. One of the B-specific partial transcripts (c_Bplus_DN27718_c0_g1_i3) encodes a protein of the OVATE family. The plant-specific proteins of this family act as transcription repressors [23]. In *A. thaliana*, overexpression of *AtOFP1* results in dwarf plants with reduced cell elongation in rapidly elongating aerial organs, including hypocotyl, leaf petiole and inflorescence stems [24]. Another example is the B-specific transcript (c_Bplus_DN30654_c0_g1_i2) encoding an IAA9/IAA20 orthologue of *A. thaliana*. The members of the Aux/IAA family are essential during plant development, such as root development, shoot growth, flower organ development and fruit ripening [25]. 

In summary, our tissue-specific RNA-seq analysis identified candidate transcripts which might be involved in B chromosome elimination in *Ae. speltoides* embryos. Tissue-specific sampling of the cellular region where B elimination occurs enabled the capturing of rare transcripts and cell-specific differences, which are otherwise hidden in whole-embryo samples. The study demonstrates that the process of B chromosome elimination and presence of B chromosomes has a surprisingly strong effect on the global transcriptome of meristematic cells in young embryos. These transcripts are either associated with the process of B chromosome elimination or activated in response to the presence of B chromosomes. The identification of B-specific candidate transcripts provides the basis for future work on the elucidation of molecular mechanisms underlying B chromosome elimination and studies validating the function of candidate genes in planta. 

## 3. Materials and Methods

### 3.1. Plant Material

*Aegilops speltoides* Tausch (PI 487238; USDA-ARS, Aberdeen, ID, USA) plants with and without B chromosomes (further referred to as Bplus and B0) were grown under greenhouse conditions (16 h light, 25 °C/19 °C day/night) at IPK, Gatersleben (Germany). At the beginning of the tillering stage, plants were kept for one month at 12 °C to ensure synchronous flowering. Presence of Bs in plants was identified by PCR with primers for the B-specific repeat AesTR-183 [26] (Appendix A) and the exact number of Bs was determined by flow-cytometric analysis according to [27]. Only plants with 2, 3 or 4 Bs were used for the generation of embryos. To ensure the presence or absence of Bs in the embryos, plants with and without Bs were kept in separate greenhouse chambers to prevent cross-pollination. The time of anthesis was recorded for each flower in the spikes. Embryos were sampled at 17 to 20 DAA, immediately frozen in liquid nitrogen and stored at −80 °C.

### 3.2. Laser Capture Microdissection (LCM) and RNA Extraction

Embryos were transferred to a cryostat (−20 °C) and then glued onto sample plates using the Tissue-Tek^®^ OCT™ compound (Sakura Finetek Europe BV, Alphen aan den Rijn, The Netherlands). Serial cross-sections of 20 µm thickness were prepared using a cryotome (Cryostar NX 70, Microm GmbH, Neuss, Germany), mounted on RNAse-free PEN membrane slides (MMI, Eching, Germany) and stored in the cryostat at −20 °C until complete dryness (5–7 days). Prior to microdissection, cryosections were adapted to room temperature for some minutes. Cells were isolated from the central meristematic root region of Bplus and B0 embryos (Figure 2), where the elimination of Bs was shown to occur [3]. LCM-based isolation using the MMI Cell Cut system, RNA isolation, mRNA amplification and RNA-seq library preparation has been performed as described in [28] with slight modifications.

Total RNA was extracted from 30–55 tissue sections for each of the three biological replicates from different genomic backgrounds using the Absolutely RNA Nanoprep Kit (Agilent Technologies, Santa Clara, CA, USA). From 3 to 4 embryos were pooled per one replicate. Number of tissue sections obtained from each embryos and total square area of the sections are given in Appendix A (Appendix A). mRNA was amplified by one round of T7-based in vitro transcription using the MessageAmp TM II aRNA Kit (InvitrogenTM) to generate 1–2 µg antisense RNA.

### 3.3. RNA Sequencing and Data Preprocessing

After quality control of RNA samples, Illumina sequencing was performed by Novogene Co. Ltd. (Hong Kong, China) using the NEBnext Ultra RNA Kit for library preparation to produce 150 bp paired-end reads. 

In total, 68.7 Gbp of paired-end reads (in average > 38million reads per sample) were generated. Prior to assembly, all reads were preprocessed for quality control with FastQC (Galaxy v0.72 [29]). After read quality inspection, the Trimmomatic program (Galaxy v0.36.6 [30]) was applied to filter out adaptors and low-quality sequences that retained 66.9 Gbp high-quality sequences (parameters: paired-end, IlluminaClip, TruSeq 3, 2:30:10, sliding window 4:15, MINLEN 36, Trailing 3; LEADING 3). Trimmed reads of replicates were combined into Bplus and B0 datasets for comparison of genomic backgrounds.

### 3.4. Sequence Assembly and Annotation

A de novo transcriptome assembly was performed with separated B0 and Bplus datasets using Trinity v2.6.5 [28] with default parameters which resulted in 214,060 and 254,246 contigs in B0 and in Bplus conditions, respectively. Trinity results provided a set of sequences (called “isoforms”) grouped into clusters. 

Quality and completeness of assemblies were determined using Transrate v1.0.3 [31], which resulted in optimized assemblies B0 and Bplus containing 165,818 and 196,360 contigs, respectively. In order to track the origin of each contig, the sequences were renamed with B0 and Bplus prefixes for each contig. In addition, B0 and Bplus files were merged into one dataset (B0&Bplus) and further processed by minimap2 v2.9 [32]. To find overlapping as well as unique contigs, a threshold of 80% identity within the aligned part of the shorter sequence was applied. The resulting FASTA files were processed by the CD-HIT-EST, v4.6.8 program (sequence identity threshold 0.90; [33,34]) to cluster highly homologous sequences and reduce redundancy. Finally, files were combined into a single file for usage in downstream applications. In particular, the final data set (117,441 contigs in total) was exploited for the detection of differentially expressed genes.

### 3.5. Quality Analysis and Differential Expression Analysis

For extraction of DEGs, the final file, including common and unique contigs in B0 and Bplus conditions (117,441 in total), was used for Kallisto indexing and read counting (v0.44.0; [35]). After that, log_2_-transformed counts-per-million (CPM) values were generated and “PtR” of the Trinity pipeline was employed to examine the data quality based on Pearson’s correlation and principal component analysis (PCA). DEG analysis was conducted using DESeq2 [36] with an FDR of 0.05 and 0.01 [37] to identify significant DEGs.

### 3.6. Transcriptome Annotation

Transdecoder v5.3.0 (http://transdecoder.github.io) was used to predict open reading frames (ORFs), which were annotated by Interproscan Galaxy v5.0.2mkh [16,17] and additionally BLASTed against the IWGSC Ref.Seq.1.0 of the wheat assembly [18] using *E*-values < 1.0E-3 in at least 100 amino acids. Completeness of transcriptomes was estimated by BUSCO (Benchmarking universal single-copy orthologs) v3 [38,39] using the plant set “Poales_odb10”.

### 3.7. GO Term Enrichment Analysis

Statistically enriched GO terms in DEGs were identified using the AgriGo v2.0 analysis toolkit [40]. Biological processes, molecular functions and cellular components were assessed using *Triticum aestivum* as a background reference and Fisher’s exact test with FDR correction. Redundant GO terms were removed using REVIGO [41] with the following parameters: medium similarity (0.7); GO categories associated to: *p*-values; GO term sizes database: *O. sativa*; semantic similarity measure to use: SimRel.

### 3.8. RT-qPCR

Residual aRNA from the RNA-seq approach was used for cDNA synthesis. First-strand cDNA was synthesized using SuperScript III (Invitrogen) with random priming according to the manufacturer’s instructions. The Power SYBR Green PCR mastermix was used to perform reactions in an ABI 7900 HT Real-Time PCR system (Applied Biosystems, Foster City, CA, USA). Data were analyzed using SDS 2.2.1 software (Applied Biosystems). Four replications were conducted for each transcript. Expression values were normalized with primers for the internal reference gene *TaGAPDH* (Appendix A) and calculated as an arithmetic mean of the replicates. Dissociation curves confirmed the presence of a single amplicon in each PCR reaction. Efficiencies of PCR reactions were determined using LinRegPCR software (http://www.gene-quantification.de/download.html). Values were calculated according [42] and given as relative expression (1+E)−ΔCt. All primers are listed in Appendix A.

### 3.9. Data Availability

Raw sequence reads can be obtained from the European Nucleotide Archive (ENA) under study accession number PRJEB39517.

## Figures and Tables

**Figure 1 ijms-21-07596-f001:**
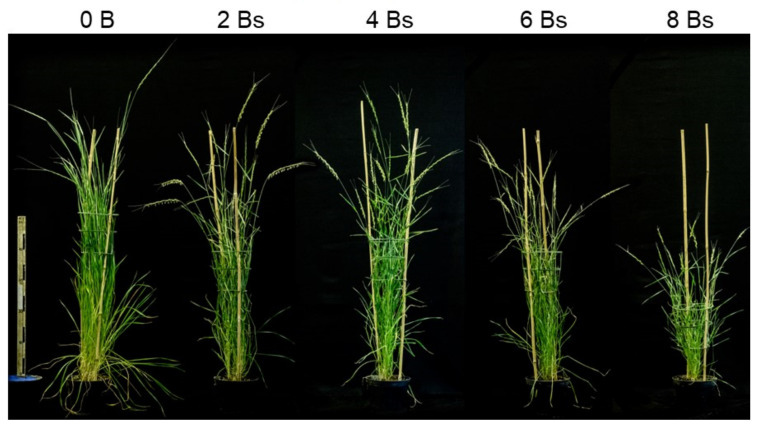
*Ae. speltoides* plants with an increasing number of B chromosomes. Vegetative growth is affected for plants owing more than 4 Bs.

**Figure 2 ijms-21-07596-f002:**
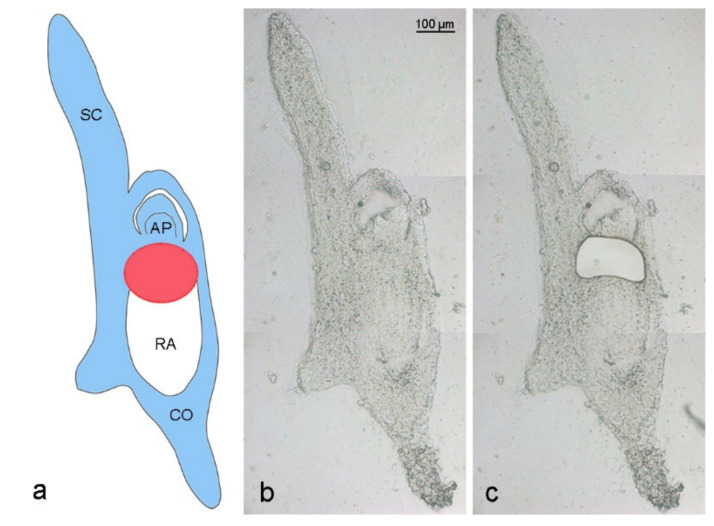
Laser capture microdissection of an *Ae. speltoides* embryo. (**a**) Schema is depicting an embryo with B chromosomes at the stage of 17 days after anthesis. Blue color indicates embryo parts containing Bs, white color shows the absence of Bs in the radicle. The red circle marks the region of ongoing Bs elimination between the apical meristem and developing radicle. SC—scutellum, AP—apical meristem, RA—radicle, CO—coleorhiza. Cryosection of a 17 DAA embryo (**b**) before and (**c**) after laser dissection of the region where Bs undergo elimination.

**Figure 3 ijms-21-07596-f003:**
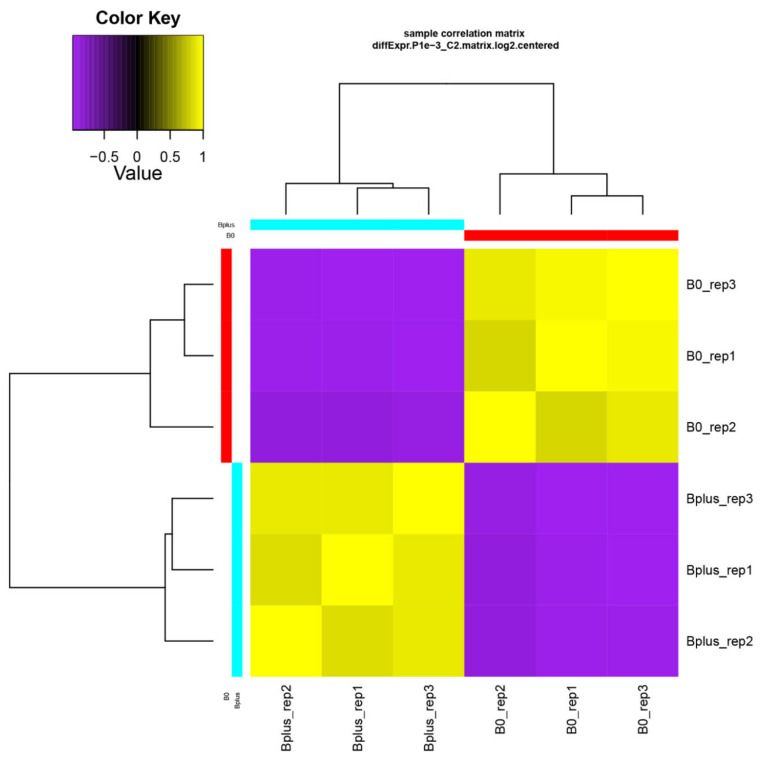
Validation of biological replicates and relationship among *Ae. speltoides* samples. The hierarchical clustering was performed following the Trinity pipeline on differentially expressed transcript isoforms (FDR ≤ 0.001; logFC ≥ 4). B0_rep1 to B0_rep3 represent biological replicates of samples without B chromosomes; Bplus_rep1 to Bplus_rep3 are samples with eliminating B chromosomes.

**Figure 4 ijms-21-07596-f004:**
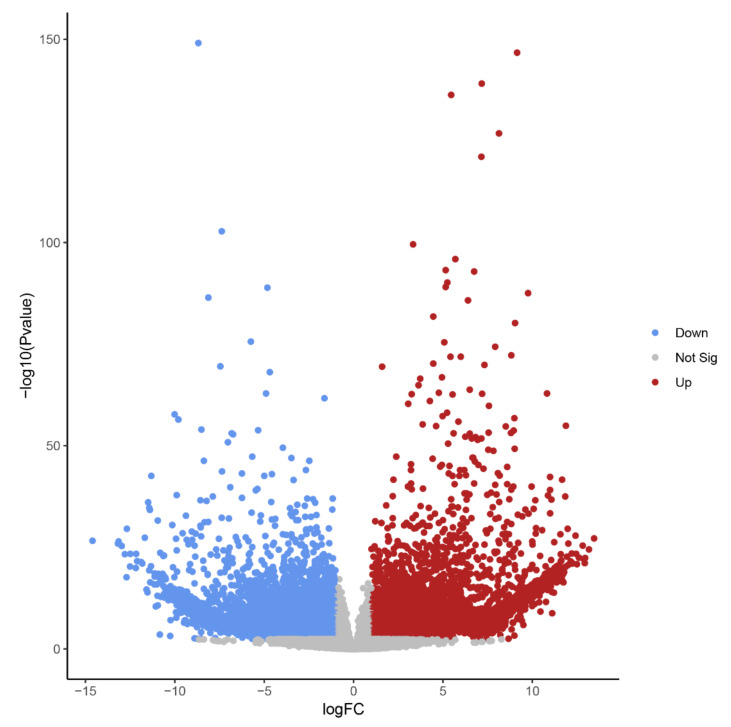
Volcano plot highlighting transcript isoforms differentially expressed during B chromosome elimination in *Ae. speltoides* (FDR-corrected *p*-value < 0.01). Red color defines up-regulated and blue defines down-regulated transcript isoforms.

**Figure 5 ijms-21-07596-f005:**
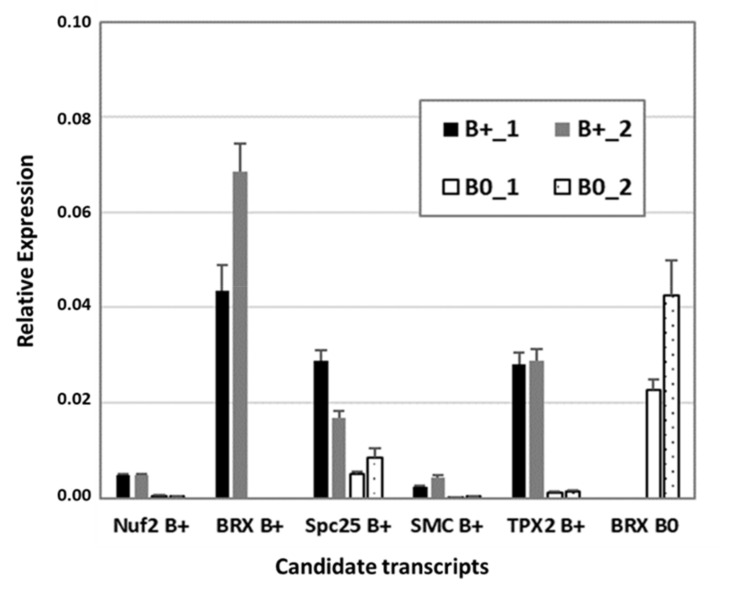
Confirmation of RNA-seq data by RT-qPCR of a subset of differentially expressed transcripts between Bplus (indicated as B+) and B0 conditions. Transcript levels were determined in two biological replicates of each genotype and normalized to GAPDH. Relative expression and standard deviations are calculated from four replicates (*n* = 4).

**Figure 6 ijms-21-07596-f006:**
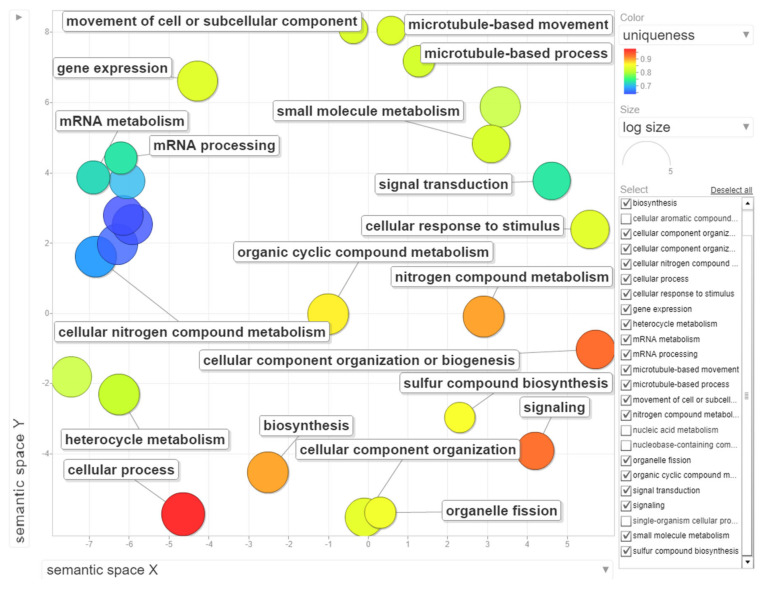
Gene ontology (GO) enrichment analysis of genes with statistically significant up-regulated changes during B chromosome elimination. REVIGO was used to summarize and visualize the enriched GO terms related to biological process. Circles depicted by filled color show significantly enriched GO terms with log_10_
*p*-value < 0.05. Similar GO terms are grouped based on semantic similarity.

**Table 1 ijms-21-07596-t001:** RNA-seq-based differential expression analysis in embryos of *Ae. speltoides* undergoing B chromosome elimination.

	Amount Studied	Differentially Expressed		Highly DE ^1^	DE B0 Unique		DE Bplus Unique	
		*p* < 0.05	*p* < 0.01	*p* < 0.01	*p* < 0.05	*p* < 0.01	*p* < 0.05	*p* < 0.01
ranscr. isoforms	41,615	21,197	16,162	14,578	390	387	245	240
Unigenes	45,533	20,276	15,044	13,519	561	524	597	519

All *p*-values represent FDR-corrected values. ^1^ Highly DE are differentially expressed (DE) candidates with FC log_2_ min value of 1.

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
