# Peer review of "Tissue-Specific Transcriptome Analysis Reveals Candidate Transcripts Associated with the Process of Programmed B Chromosome Elimination in Aegilops speltoides"

_ijms, 2020, doi:10.3390/ijms21207596_

Round 1
Reviewer 1 Report
The manuscript reports on the tissue-specific transcriptome analysis of the embryos of Aegilops speltoides during programmed B chromosomes elimination. The RNA-seq analysis was conducted for central meristematic regions of plants with and without B chromosomes. The tissues were isolated using laser capture microdissection. Among 41,615 analyzed transcript isoforms, 35% transcript isoforms, which differentially expressed during B chromosomes elimination, were identified. Up-regulated and down-regulated genes were classified in plants with B chromosomes. B-unique transcript isoforms were identified, and 70 were functionally annotated. The candidate genes associated with elimination of accessory chromosomes were categorized. To date, this is the first research on comparative tissue-specific transcriptome analyses of programmed chromosome elimination.
I recommend acceptance in the present form.
Few corrections need to be addressed.
In the title, I suggest specifying programmed B chromosome elimination.
Line 7: and Andreas Houben
In the Supplementary data zip file, Fig.S2 is mistakenly duplicated, and Fig.S3 does not exist.
Author Response
CC
Reviewer 1
The manuscript reports on the tissue-specific transcriptome analysis of the embryos of Aegilops speltoides during programmed B chromosomes elimination. The RNA-seq analysis was conducted for central meristematic regions of plants with and without B chromosomes. The tissues were isolated using laser capture microdissection. Among 41,615 analyzed transcript isoforms, 35% transcript isoforms, which differentially expressed during B chromosomes elimination, were identified. Up-regulated and down-regulated genes were classified in plants with B chromosomes. B-unique transcript isoforms were identified, and 70 were functionally annotated. The candidate genes associated with elimination of accessory chromosomes were categorized. To date, this is the first research on comparative tissue-specific transcriptome analyses of programmed chromosome elimination.
I recommend acceptance in the present form.
RESPONSE:
Many thanks for your kind comments.
Few corrections need to be addressed.
In the title, I suggest specifying programmed B chromosome elimination.
RESPONSE:
Many thanks for your suggestions. The title has been changed to: Tissue-specific transcriptome analysis reveals candidate transcripts associated with the process of programmed B chromosome elimination in Aegilops speltoides
Line 7: and Andreas Houben
RESPONSE:
Corrected
In the Supplementary data zip file, Fig.S2 is mistakenly duplicated, and Fig.S3 does not exist.
RESPONSE:
Corrected
Reviewer 2 Report
In the manuscript “Tissue-specific transcriptome analysis reveals candidate genes associated with the process of programmed chromosome elimination in Aegilops speltoides” the authors analyzed transcription in the central meristematic region of Aegilops speltoides embryos where B chromosome elimination occurs. Performed analysis was based on transcriptome comparison between central meristematic regions isolated from plants with (Bplus) and without (B0) B chromosomes (Bs). Isolation of the region of interest was performed by laser capture microdissection and the comparative RNA-seq analysis of meristematic cells of embryos with and without Bs revealed numerous differentially expressed transcript isoforms. Annotated unigenes were found to be up-regulated and to be down-regulated in Bplus condition. No doubt, that obtained results are very interesting and important. However, I have some general question and comments.
- The comparative RNA-seq analysis was performed between transcriptome of cells from the region in which ‘programmed B chromosome elimination take place’ in the Bplus and B0 plants. Cells were isolated in time when ‘programmed B chromosome elimination’ occured. The author suppose that at least part of revealed distinctions are associated with the B chromosome elimination. Probably, it is true. However, some of these distinctions could be associated just with the presence of Bs in some of isolated cells. I suppose that comparison of transcriptomes of the cells isolated from the same tissue of the plant with and without Bs should be also carried out.
At least, the authors should explain, why such comparison could be omitted.
- Are there available transcriptome data on plant tissues with and without Bs in speltoides?
- Number of the Bs can varied in speltoides plants from 0 to 8. However, the authors divided all plants into two groups, plants with and without Bs. I understand that estimation of B chromosome number in the embryos on that stage is difficult. However, we should remember that plants with low B chromosome number show almost normal phenotype while plants with 8 B chromosomes are much smaller in size. It makes us to expect that transcriptome of the cells from the Bplus and B0 plants could be different. The authors should discussed at least this problem. Could the observed difference between transcriptomes of isolated cells derived from the presence of the Bs or even from the different number of the Bs in the cells? Could the observed difference between transcriptomes of isolated cells derived partly from the presence of the Bs and partly could be associated with the B chromosome elimination on that time?
- The authors wrote: ‘In summary, our tissue-specific RNA-seq analysis identified B-specific candidate genes which might be crucial for B chromosome elimination in speltoides embryos’. I agree that these genes MIGHT be crucial for B chromosome elimination in Ae. speltoides embryos. However, they could be not involved in B chromosome elimination at all. I suppose that without the comparative RNA-seq analysis of cells from tissues in plants with and without Bs conclusion that ‘our tissue-specific RNA-seq analysis identified B-specific candidate genes which might be crucial for B chromosome elimination in Ae. speltoides embryos’cannot be done.
- The authors applied the terms ‘B-specific candidate genes’ and ‘B chromosome specific transcript isoforms’. They should explain what it means.
- The authors made statement that ‘The study demonstrates that the process of B chromosome elimination has a surprisingly strong effect on the global transcriptome of meristematic cells in young embryos’. The authors should explain how they differentiate effects of B chromosome elimination and B chromosome presence.
- All conventional signs in the Figures should be explained in legends. For example on the Figure 3 there are “B0_rep1, Bplus_rep1” and so on without explanation.
- The author list should be complete. In the manuscript it is: ‘Anastassia Boudichevskaia1, 2, Alevtina Ruban1, 2, Johannes Thiel1, Anne Fiebig1, Andreas Houben1 and *’
The manuscript could be published in the ‘IJMS’ after major revision.
Author Response
Reviewer 2
In the manuscript “Tissue-specific transcriptome analysis reveals candidate genes associated with the process of programmed chromosome elimination in Aegilops speltoides” the authors analyzed transcription in the central meristematic region of Aegilops speltoides embryos where B chromosome elimination occurs. Performed analysis was based on transcriptome comparison between central meristematic regions isolated from plants with (Bplus) and without (B0) B chromosomes (Bs). Isolation of the region of interest was performed by laser capture microdissection and the comparative RNA-seq analysis of meristematic cells of embryos with and without Bs revealed numerous differentially expressed transcript isoforms. Annotated unigenes were found to be up-regulated and to be down-regulated in Bplus condition. No doubt, that obtained results are very interesting and important. However, I have some general question and comments.
The comparative RNA-seq analysis was performed between transcriptome of cells from the region in which ‘programmed B chromosome elimination take place’ in the Bplus and B0 plants. Cells were isolated in time when ‘programmed B chromosome elimination’ occured. The author suppose that at least part of revealed distinctions are associated with the B chromosome elimination. Probably, it is true. However, some of these distinctions could be associated just with the presence of Bs in some of isolated cells. I suppose that comparison of transcriptomes of the cells isolated from the same tissue of the plant with and without Bs should be also carried out.
At least, the authors should explain, why such comparison could be omitted.
RESPONSE
Tissue-specific sampling of the cellular region where B chromosome elimination occurs enabled capturing of rare transcripts and cell-specific differences, which are otherwise hidden in whole-embryo samples. We agree that the comparison of transcriptomes of tissues with and without Bs (+B/0B) is of interest. Based on our previous experiments with rye (Banaei-Moghaddam et al., 2013; Ma et al., 2017) and other species (reviewed in (Benetta et al., 2019)) it is demonstrated that Bs are a carrier of genic sequences which are active in a tissue-specific way. They can even affect the transcriptome of the host genome in trans. In Ae. speltoides we aimed to compare the transcriptome of the embryonic regions where the elimination of Bs occurs. The ideal experiment would be to compare the transcriptome of the same embryonic regions without B, with Bs and Bs undergoing elimination. However, unfortunately, this experiment is not possible as Bs always undergo elimination in this region. The comparison with other tissues of the plant where Bs are stably present or absent is less informative, as B-located genes are active in a tissue-type specific way. For that reason, we did not perform the suggested experiment, but added in the text the information that Bs itself also form transcripts irrespectively of the process of B chromosome elimination. In INTRODUCTION: “Analysis of different species demonstrated that Bs encode active genic sequences and also influence the transcriptome of standard genomes [reviewed by 5].”
- Are there available transcriptome data on plant tissues with and without Bs in speltoides?
RESPONSE:
Unfortunately, no additional transcriptome data of other Ae. speltoides (0B/+B) tissues are available.
- Number of the Bs can varied in speltoides plants from 0 to 8. However, the authors divided all plants into two groups, plants with and without Bs. I understand that estimation of B chromosome number in the embryos on that stage is difficult. However, we should remember that plants with low B chromosome number show almost normal phenotype while plants with 8 B chromosomes are much smaller in size. It makes us to expect that transcriptome of the cells from the Bplus and B0 plants could be different. The authors should discussed at least this problem. Could the observed difference between transcriptomes of isolated cells derived from the presence of the Bs or even from the different number of the Bs in the cells? Could the observed difference between transcriptomes of isolated cells derived partly from the presence of the Bs and partly could be associated with the B chromosome elimination on that time?
RESPONSE:
We agree that the number of Bs could influence the analysed transcriptome. However, as the process of B chromosome elimination works irrespectively of the number of Bs and as there is no straightforward way of determining exact B chromosome number in embryos, we thought appropriate to pool embryos together in each replicate. Somehow this information was missing in Material and Methods and is now corrected. Following sentencey was added, including Table S9 with detailed information for the number of sections: “From 3 to 4 embryos were pooled per one replicate. Number of tissue sections obtained from each embryos and total square area of the sections are given in the Table S9 (Supplementary Materials).”
and in Material and Methods
“Presence of Bs in plants was identified by PCR with primers for the B-specific repeat AesTR-183 [11] (Table S1, Supplementary Materials) and exact number of Bs was determined by flow-cytometric analysis according to (Ruban et al. 2014). Only plants with 2, 3 or 4 Bs were used for the generation of embryos.
- The authors wrote: ‘In summary, our tissue-specific RNA-seq analysis identified B-specific candidate genes which might be crucial for B chromosome elimination in speltoides embryos’. I agree that these genes MIGHT be crucial for B chromosome elimination in Ae. speltoides embryos. However, they could be not involved in B chromosome elimination at all. I suppose that without the comparative RNA-seq analysis of cells from tissues in plants with and without Bs conclusion that ‘our tissue-specific RNA-seq analysis identified B-specific candidate genes which might be crucial for B chromosome elimination in Ae. speltoides embryos’cannot be done.
RESPONSE:
We agree that additional work will be required to identify genes truly involved in the process of chromosome elimination. In order tone down our conclusions we have added and adjusted the following sentences.
ABSTRACT “These genes are either associated with the process of B chromosome elimination, transcribed by Bs irrespectively of the elimination process or transcribed by the standard A chromosomes in response to the presence Bs”.
INTRODUCTION: we added: “Analysis of different species demonstrated that Bs encode active genic sequences and also influence the transcriptome of the standard genome [reviewed by 5].”
RESULTS/DISCUSSION: “The high number of differentially regulated transcript isoforms (35% out of 41,615 analyzed transcript isoforms) indicates that the process of B chromosome elimination and the presence of B chromosomes have a strong effect on the transcriptome of the embryonic central meristematic region (Figure 4).”
“Transcript isoforms detected during chromosome elimination represented several categories: transposons (such as transposase, retrotransposon); potential transposons (zinc finger); unknown; and genes with specified functions.”
Last Paragraph: “The study demonstrates that the process of B chromosome elimination and presence of B chromosomes has a surprisingly strong effect on the global transcriptome of meristematic cells in young embryos. These transcripts are either associated with the process of B chromosome elimination, transcribed by Bs irrespectively of the elimination process or transcribed by the standard A chromosomes in response to the presence Bs. The identification of B-specific candidate genes provides the basis for future work on the elucidation of molecular mechanisms underlying B chromosome elimination and studies validating the function of candidate genes in planta.
- The authors applied the terms ‘B-specific candidate genes’ and ‘B chromosome specific transcript isoforms’. They should explain what it means.
RESPONSE:
We have added to the manuscript following explonation: “De novo transcriptome assembly based on Trinity was applied since Ae. speltoides represents a non-model organism that contains in addition a high proportion of heterozygous loci. Trinity is highly effective for reconstructing transcripts and alternative spliced isoforms (Grabherr et al 2011) and it allowed a better understanding which transcript isoforms are truly expressed under specific condition such as elimination of Bs”.
As mentioned in Material and Methods, Trinity results provided a set of sequences (called “isoforms”) grouped into clusters. In some papers, Trinity-based transcript isoforms are called transcripts (an example is the paper of Christie et al (2017), https://www.ncbi.nlm.nih.gov/pmc/articles/PMC5796769/) who applied Trinity for the tissue-specific RNA-sequencing (see Table 4, isoforms are called transcripts).
Under candidate genes we mean truly expressed isoforms that might shed light on the mechanism of B chromosome elimination. Accurate isoform identification can assist researches in further functional analyses. We agree and replaced in the text “candidate genes” with “candidate transcripts”.
- The authors made statement that ‘The study demonstrates that the process of B chromosome elimination has a surprisingly strong effect on the global transcriptome of meristematic cells in young embryos’. The authors should explain how they differentiate effects of B chromosome elimination and B chromosome presence.
RESPONSE:
In order to tone down this statement we extend the sentence in the following way: “Tissue-specific RNA-seq revealed that the elimination of Bs has a strong effect on the transcriptome of the meristematic central zone of young embryos. In particular, 35% from 41,615 analyzed transcript isoforms showed statistically significant altered expression during the elimination process. These transcripts are either associated with the process of B chromosome elimination, transcribed by Bs irrespectively of the elimination process or transcribed by the standard A chromosomes in response to the presence Bs.”
- All conventional signs in the Figures should be explained in legends. For example on the Figure 3 there are “B0_rep1, Bplus_rep1” and so on without explanation.
RESPONSE
(added information is labeled in yellow):
Figure 3. Validation of biological replicates and relationship among Ae. speltoides samples (B0 and Bplus conditions). The hierarchical clustering was performed following the Trinity pipeline on differentially expressed transcript isoforms (FDR ≤ 0.001; logFC ≥ 4). B0_rep1 to B0_rep3 represent biological replicates of samples without B chromosomes; Bplus_rep1 to Bplus_rep3 are samples with eliminating B chromosomes.
Figure 5. Confirmation of RNA-seq data by RT-qPCR of a subset of differentially expressed genes (DEGs) between Bplus (indicated as B+) and B0 conditions. Transcript levels were determined in two biological replicates of each genotype and normalized to GAPDH. Relative expression and standard deviations are calculated from four replicates (n = 4).
Figure S1. Assessing variability of biological replicates and relationship among samples without B chromosomes (B0) and with eliminating B chromosomes (Bplus). Principal Component Analysis (PCA) plots displaying variability within the 6 samples along PC1 and PC2 as well as PC2 and PC3.
- The author list should be complete. In the manuscript it is: ‘Anastassia Boudichevskaia1, 2, Alevtina Ruban1, 2, Johannes Thiel1, Anne Fiebig1, Andreas Houben1 and *’
RESPONSE: corrected
Banaei-Moghaddam, A.M., Meier, K., Karimi-Ashtiyani, R., and Houben, A. (2013). Formation and expression of pseudogenes on the B chromosome of rye. Plant Cell 25, 2536-2544.
Benetta, E.D., Akbari, O.S., and Ferree, P.M. (2019). Sequence expression of supernumerary B chromosomes: Function or fluff? Genes 10, 123.
Ma, W., Gabriel, T.S., Martis, M.M., Gursinsky, T., Schubert, V., Vrana, J., Dolezel, J., Grundlach, H., Altschmied, L., Scholz, U., et al. (2017). Rye B chromosomes encode a functional Argonaute-like protein with in vitro slicer activities similar to its A chromosome paralog. New Phytol 213, 916-928.
Round 2
Reviewer 2 Report
The manuscript can be accepted in present form.